# Lower Genital Tract Microbiome in Early Pregnancy in the Eastern European Population

**DOI:** 10.3390/microorganisms10122368

**Published:** 2022-11-30

**Authors:** Mariya Gryaznova, Olga Lebedeva, Olesya Kozarenko, Yuliya Smirnova, Inna Burakova, Mikhail Syromyatnikov, Alexander Maslov, Vasily Popov

**Affiliations:** 1Laboratory of Metagenomics and Food Biotechnology, Voronezh State University of Engineering Technologies, 394036 Voronezh, Russia; 2Department of Genetics, Cytology and Bioengineering, Voronezh State University, 394018 Voronezh, Russia; 3Department of Obstetrics and Gynecology, Belgorod State National University, 308015 Belgorod, Russia; 4Antenatal Care Department, Yakovlevo Central District Hospital, 309070 Stroitel, Russia; 5Department of Genetics, Albert Einstein College of Medicine, New York, NY 10461, USA

**Keywords:** cervical microbiome, vaginal microbiome, early pregnancy, first trimester, sequencing, female genital tract, *16S rRNA*, *Lactobacillus inners*, NGS

## Abstract

Background: It is known that the features of the cervicovaginal microbiome can depend on ethnicity, which might be caused by genetic factors, as well as differences in diet and lifestyle. There is no research on the cervicovaginal microbiome of Eastern European women during early pregnancy. Methods: We evaluated the cervical and cervicovaginal microbiome of women with first-trimester pregnancy (*n* = 22), further delivered at term, using the *16S rRNA* sequencing method. Results: The predominant bacterial species in both groups was *Lactobacillus iners*, followed by *Prevotella copri*, *Ileibacterium valens*, *Gardnerella vaginalis* and *Muribaculum intestinale* in the cervical samples, and *Gardnerella vaginalis*, *Prevotella copri*, *Bifidobacterium longum*, *Ileibacterium valens* and *Muribaculum intestinale* in the cervicovaginal samples. The cervical microbiome had higher alpha diversity; a higher abundance of *Muribaculum intestinale, Aquabacterium parvum* and *Methyloversatilis universalis*; and a lower abundance of *Psychrobacillus psychrodurans*. Conclusions: The *Lactobacillus iners*-dominated microbiome (CST III) was the predominant type of cervical and cervicovaginal microbiome in early pregnancy in the majority of the women. The presence of soil and animal bacteria in the cervicovaginal microbiome can be explained by the rural origin of patients.

## 1. Introduction

The microbiome of the lower genital tract in early pregnancy is not well understood, despite its critical role in the course and outcome of pregnancy.

It is known that the structure of the cervicovaginal microbiome can depend on ethnicity [1,2,3,4]. In a review by Gupta et al. (2017), who summarized these data, it was shown that lactobacillus-dominated microbiome predominates in Caucasian and Asian women, while anaerobic microorganisms associated with bacterial vaginosis predominate in African and Hispanic women [5]. These ethnic differences are observed regardless of the region of residence of the patients [6]. As it is known that microbiome shaping can be under the influence of local immunity [7], including Toll-like and NOD-like receptors [8,9,10], and antimicrobial peptides [11], it was suggested that they occur due to genetic factors, including the characteristics of the genes responsible for the systems of innate and acquired immunity, and not sexual behaviour, diet, or preferred methods of contraception [5].

Ethnic differences in the vaginal microbiome also were described in the first trimester of pregnancy [12,13,14]. But, to our knowledge, there is no research on the normal cervicovaginal microbiome of Eastern European women during early pregnancy. Therefore, this study aims to evaluate the cervicovaginal microbiome of Eastern European women during early pregnancy in healthy patients, who later have labour at term.

## 2. Materials and Methods

### 2.1. Ethics and Patient Consent Guidelines

All patients gave written agreement to the use of their biomaterial, as well as anonymized anamnesis data for this study of the microbiome landscape of the female genital tract. The study was approved by the Ethical Committee of Voronezh State University (protocol No. 42-05 of 27 December 2021).

### 2.2. Study Design

In a prospective longitudinal cohort study, healthy women (*n* = 24) with singleton pregnancies were recruited at the beginning of their antenatal care (8–11 weeks of gestation) after obtaining their written informed consent. Among them, only patients with normal pregnancy, delivered at term (37 to 42 weeks of gestation) (*n* = 22), were included in the study. See Appendix A for complete patient information. Antenatal care was provided in the outpatient clinic of the Yakovlevo central district hospital in the rural area (Belgorod region, Russia). The ethnicity of all patients was Russian.

Exclusion criteria included the usage of any prebiotics, probiotics, antibiotics or synbiotics during the previous month; vaginal intercourse during the last 3 days; multiple pregnancies; vaginal bleeding; presence of severe non-obstetrical conditions, including primary and secondary immune deficiency; and refusal to sign the informed consent form or refusal to be followed up. Basic clinical characteristics are shown in Table 1.

To study the microbiome of the female genital tract, the material was collected using a cytobrush (cervical canal) and a vaginal swab (vagina). The biomaterial was collected in two Eppendorf tubes (the 1st—from the cervical canal by cytobrush; the 2nd—from the cervical canal by cytobrush and the vagina by vaginal swab) and mixed with RNAlater™ stabilization solution (Thermo Fisher Scientific, Madison, WI, USA) and then delivered to the laboratory at −4 to −8 °C. In the laboratory, the biomaterial was stored at a temperature of −80 °C before the start of the investigation.

### 2.3. DNA Extraction and 16S rRNA Amplification

For DNA extraction, we used the ZymoBIOMICS DNA Miniprep Kit by the manufacturing protocol (Zymo Research, Los Angeles, CA, USA). In this step of the experiment, we added an extra sample that contained Milli-Q water used in the laboratory and a cropped sterile cytobrush lowered into it. This sample went through all the same stages of sample preparation as the samples under study. This was necessary to eliminate contaminants from the test samples at the data processing stage. For quality control, we performed in the 1.5% agarose gel; and for Quantity control, we used a Qubit 2.0 fluorometer with a dsDNA HS Assay Kit (Thermo Fisher Scientific, Madison, WI, USA). All DNA samples were applicable for further analysis. 

We chose hypervariable region V3 of the *16S rRNA* gene for performing the microbiome study. For targeting amplification, this region universal primers set of 337F (5′-GACTCCTACGGGAGGCWGCAG-3′) and 518R (5′-GTATTACCGCGGCTGCTGG-3′) and a 5X ScreenMix-HS Master Mix Kit (Evrogen, Moscow, Russia) were used. Amplification was performed with the Bio-Rad CFX96 Touch™ Real-Time PCR Detection System (Bio-Rad, Hercules, CA, USA) according to the next protocol: total denaturation—94 °C for 4 min; 35 cycles of denaturation—94 °C for 30 s; primer annealing—53 °C for 30 s and elongation—72 °C for 30 s; and the final elongation—72 °C for 5 min.

### 2.4. Library Preparation and Sequencing

The amplicons were purified with AMPureXP magnetic beads (Beckman Coulter, Brea, CA, USA) following the protocol, before the preparation of libraries for sequencing on Ion Torrent PGM. For the library preparation, we used the NEBNext Fast DNA kit (New England Biolabs, Ipswich, MA, USA) according to the manufacturer’s instructions. Barcoding was performed with the NEXTflex kit (Ion Torrent; 64 adapters; PerkinElmer, Inc., Waltham, MA, USA). After the adapters ligation, the purification was conducted using AMPureXP magnetic beads (Beckman Coulter, Brea, CA, USA) again. The last step of library preparation was completing the quantification of the libraries using the KAPA Library Quantification Kit for Ion Torrent Platforms (F. Hoffmann-La Roche Ltd., Basel, Switzerland). All libraries had high quality and were applicable for sequencing.

For conducting an emulsion PCR, preparing and downloading the chip, and adjusting the Ion Torrent Sequencer, we used the Ion PGM Hi-Q View OT2 Kit and the Ion PGM Hi-Q View Sequencing Kit according to protocols (ThermoFisher Scientific, Madison, WI, USA). Libraries loading was performed on the Ion 318 Chip v2 BC chip.

### 2.5. Processing of the Sequencing Data

Preprocessing was performed with Torrent Suite Software for base calling and alignment. The reads in BAM format were converted into FASTQ format with the FileExporter plugin for easy downstream analysis with R programming tools. Raw sequencing data were available from the NCBI BioProject database (BioProjectID: PRJNA886610).

All of the next bioinformatics analysis was performed in RStudio (version 1.1.414 RStudio Inc., RStudio PBC, Boston, MA, USA). 

Poor quality reads were filtered out from subsequent analysis using the maximum expected error threshold of 1.0 (DADA2 package). At the next stage, the filtered reads of suitable quality were unified in length and demultiplexed. Then, during the dereplication, all identical reads were combined into unique sequences (ASV). To form operational taxonomic units (OTUs), we used the UNOISE2 algorithm.

The general identification of the taxonomic composition of microorganisms was carried out based on the SILVA database version 138 (https://www.arb-silva.de/, accessed on 26 August 2022). To assign taxonomic status, we used a relative match limit with sequence variants of 97%. We excluded contamination agents from samples with the decontam R package. Full information on all obtained reads for each test sample is contained in Appendix A.

### 2.6. Statistical Analysis

The statistical analyses were completed with GraphPad Prism 9 software (GraphPad, San Diego, CA, USA). We checked the distribution of our data with the Kolmogorov–Smirnov normality test and detected that it is not normal. Consequently, all of the following test criteria were based on this fact about the distribution. A comparative analysis of the microbiome composition from the two studied areas of the female genital tract was performed using the Multiple Wilcoxon test. Correlation analysis was carried out using Spearman’s correlation criteria. Correlations were considered very strong if R ≥ |0.8| and *p* ≤ 0.05 [15]. Alpha diversity was determined using the Shannon index. Results are presented as median (IQR, 25th–75th percentile).

## 3. Results

After processing the sequencing data, we obtained 130,842 unique reads from the studied samples, which correspond to 82 bacterial species (Figure 1). 

Despite a large number of detected species, only six bacterial species were identified in the cervical and cervicovaginal microbiome, which forms the basis of the microbiome in the studied groups. All other bacteria were grouped under “Other” (Figure 2). 

Thus, the predominant bacterial species in both groups was *Lactobacillus iners*: its median abundance in the cervical microbiome group was 0.5987 (0.1359–0.9340), and in the cervical canal and vagina group was 0.7898 (0.3201–0.9343). The next most abundant species in the cervical canal group were *Prevotella copri*—0.0269 (0.0016–0.0635), *Bifidobacterium longum*—0.0082 (0.0000–0.0843), *Ileibacterium valens*—0.0080 (0.0003–0.0551), *Gardnerella vaginalis*—0.0050 (0.0001–0.0448) and *Muribaculum intestinale*—0.0050 (0.0009–0.0206). In the group of the cervical canal and vagina, the distribution was different. The next after *Lactobacillus iners* in terms of abundance was the species *Gardnerella vaginalis*—0.0186 (0.0000–0.0449), then the species *Prevotella copri*—0.0145 (0.0006–0.0274), *Bifidobacterium longum*—0.0021 (0.0000–0.0250), *Ileibacterium valens*—0.0020 (0.0000–0.0440) and *Muribaculum intestinale*—0.0002 (0.0000–0.0130).

We also evaluated the alpha diversity in both study groups using the Shannon index. Wilkson’s nonparametric rank sum test was used to assess differences in alpha diversity between groups (Figure 3).

According to the results of the analysis, the Shannon index for the cervical canal group was 0.9216, while for the cervical canal and vagina group this indicator was 0.8268. The alpha microbiome diversity of the cervical microbiome was statistically significantly higher than in the group where cervical and vaginal samples were mixed (*p* = 0.0181).

Differences in microbiome composition between the two study groups were assessed for all detected bacterial species. Thus, according to the data obtained, statistically–significant differences were observed for the species *Muribaculum intestinale* (*p* = 0.0029)*, Aquabacterium parvum* (*p* = 0.0156) and *Methyloversatilis universalis* (*p* = 0.0156), which prevailed in the samples taken only from the cervical canal, while *Psychrobacillus psychrodurans* (*p* = 0.0154) prevailed in the group where the samples were taken from the cervical canal and vagina.

In total, we found 32 very strong correlations (R ≥ |0.8| and *p* ≤ 0.05) between bacteria in the cervical canal and 22—in cervicovaginal samples (Figure 4 and Figure 5). In the text, we describe only Spearman’s correlations with R ≥ |0.9| and *p* ≤ 0.05. 

Spearman’s correlation analysis showed that in the cervical canal group, there is a complete (R = 1) direct correlation between the following microorganisms: *Nocardioides dilutus* and *Deinococcus aerolatus*, *Porphyromonas bennonis* and *Erysipelotrichaceae UCG-003 bacterium*, *Mucispirillum schaedleri* and *Caulobacter henricii* (*p* = 0 in all cases). We also observed a strong correlation in this group between *Bacteroides graminisolvens* and *Bacteroides stercoris* (R = 0.9442; *p* = 4.20 × 10^−11^); *Pseudomonas aeruginosa* correlated with *Dubosiella newyorkensis* (R = 0.9901; *p* = 1.58 × 10^−18^) and *Alistipes putredinis* (R = 0.9950; *p* = 1.57 × 10^−21^), which also correlated with each other (R = 0.9802; *p* = 1.55 × 10^−15^) (Figure 4).

At the same time, in the cervical and vaginal group, we observed a complete correlation between Phascolarctobacterium succinatutens and Ruminococcus bromii, Turicibacter sanguinis and Sutterella wadsworthensis, Rheinheimera aquimaris and Schlegelella aquatica, and between Sphingobacterium spiritivorum and Bifidobacterium breve (*p* = 0 in all cases). We also observed a strong correlation between Dubosiella newyorkensis and Curvibacter fontanus (R = 0.9141; *p* = 2.75 × 10^−9^), as well as Bacteroides uniformis (R = 0.9958; *p* = 2.95 × 10^−22^), which also correlated with each other (R = 0.9141; *p* = 2.75 × 10^−9^) (Figure 5).

## 4. Discussion

In this study, we described for the first time the lower genital tract microbiome, including the microbiome of the cervical canal, in Eastern European women in early pregnancy. In the cervical microbiome, as well as in its mixture with the vaginal microbiome, the predominant species was *Lactobacillus iners*. According to the classification of France et al. (2020), this type of microbiome is called community state type III (CST III) [16]. To our knowledge, there are only a few studies of the cervical microbiome during the first trimester of pregnancy, and none of them used samples taken by the cytobrush to detect intracellular microorganisms. In the study of the cervicovaginal microbiome in women of Hispanic ancestry, CST III (*L. iners*) was also found as the dominating community, followed by CST IV. Samples were taken by vaginal swab from ecto– and endocervix in early pregnancy and then mixed [17]. The limitation of the study was the small number of samples (*n* = 10). Another study included separate cervical and vaginal samples taken by swab in women of Chinese origin [18]. It was found that the dominating community during early pregnancy was also CST III, followed by CST II and CST I. No differences were found between the microbiomes of the cervix and the vagina and their alpha diversity. The limitation of the study was the small number of samples (*n* = 6).

Unlike previous authors, we found differences in the abundance of a few microorganisms between cervical and cervicovaginal samples. *Muribaculum intestinale*, *Aquabacterium parvum* and *Methyloversatilis universalis* prevailed in samples taken from the cervical canal, while *Psychrobacillus psychrodurans* prevailed in the mixed cervicovaginal samples. *Muribaculum intestinale* is a strictly anaerobic bacterium, mostly found in the intestine of rodents [19,20]. But in our study, this microorganism was found in both cervical and cervicovaginal samples. To our knowledge, there is no previous data about the presence of this microorganism in the genital tract of humans. *Aquabacterium parvum* is Gram–negative aerobic bacteria, previously found in the vaginal microbiome [21], with its higher abundance in advanced maternal-age women (>35 years old) [22]. *Methyloversatilis universalis* was found in head and neck squamous cell carcinoma of patients with poor prognosis [23,24]. No data about its presence in the genital tract was found in the literature. *Psychrobacillus psychrodurans* is a soil bacteria. No data about its presence in the genital tract was found. Its higher abundance in cervicovaginal samples compared to cervical can be explained by external contamination, as all patients are residents of rural areas. In particular, these patients live in private households and grow fruits and vegetables for their consumption. None of them works on the farm. The patients have constant contact with domestic animals, mainly dogs and cats. They also have cellars or basements, where they store vegetables and fruits grown on their own or purchased in a store, where they might be contaminated by the murine microbiome. Further research is needed to define the microbiome differences between urban and rural residents, who have constant contact with soil, as well as wild and domestic animals. 

Our results also confirm the data previously received by Chen et al. (2017) for non-pregnant women: the cervical microbiome has similarity to the vaginal one, but with higher diversity and lower abundance of the *Lactobacillus* spp. [25].

The role of ethnicity in the shaping of the genital tract microbiome was discussed only in the studies on the vaginal microbiome.

The predominance of *L. iners* in the vaginal microbiome was observed in the Chinese population during the first trimester of pregnancy [12]. Two other abundant species in both populations included *Gardnerella vaginalis* and *Prevotella*, while other species in the microbiome composition were different compared to our research group. The limitation of the study was the usage of the qPCR method with a limited number of detected microorganisms.

In the African–American population, the predominant type of vaginal microbiome during the first half of pregnancy was CST III (*L. iners*, 51.8%), while CST V was not found [26].

Another study of the United States population, which included 613 pregnant women, showed the prevalence of *L. iners* in the vaginal microbiome in all ethnic groups [27], which was even significantly higher in the African ancestry group. It was coupled with a lower abundance of *G. vaginalis* vagitype and higher diversity in women of African ancestry compared to European and Hispanic ancestry. The authors emphasized that the causes of these differences remain unclear. African and Hispanic women in the research had significantly younger ages and lower household incomes compared to the European women, but previously in the non–pregnant cohort, the same authors showed that ethnicity has a stronger association with the diversity of the microbiome than socioeconomic status [6].

In the Canadian population, the predominant vaginal communities in early pregnancy were *Lactobacillus*–dominated CSTs: CST I (*L. crispatus*–dominated), followed by CST III (*L. iners*–dominated), CST V (*L. jensenii*–dominated) and CST II (*L. gasseri*–dominated) [14]. CST IV included mostly *Gardnerella vaginalis*–dominated or *Bifidobacterium*–dominated types of the microbiome. The study group included 64% of White (Caucasian) women, 21% of Asian women and 15% of women of other ethnicities. The authors did not find any relationship between CSTs and ethnicity. 

In the British population in early pregnancy, the most commonly observed vaginal community was CST I (*L. crispatus*, 40%), followed by CST III (*L. iners*, 30%), CST V (*L. jensenii*, 13%) and CST II (*L. gasseri*, 9%) [13]. The lowest diversity was observed in CST I samples, where *L. crispatus* was the only species in the community in 85% of cases, with a small amount of other *Lactobacillus* species and *Prevotella* in the rest of the samples. CSTs I, III and IV were equally distributed among White, Asian and Black ethnicities. CST II (*L. gasseri*) was not found, and CST V (*L. jensenii*) was rarely found in samples collected from Black women. The authors concluded that ethnic and geographical differences may play an important role in the structure of the vaginal microbiome during pregnancy.

Meanwhile, to our knowledge, there are no studies explaining the role of genetic factors, such as gene polymorphisms, as well as features of the lifestyle, reproductive behaviour, and diet in different ethnicities, which might explain differences in CSTs during early pregnancy.

It is known that vaginal bacteria can have antagonistic or symbiotic relationships with each other [28]. These relationships are based on the production of lactic acid, enzymes and other substances, as well as biofilm formation and the phenomenon of quorum sensing in bacterial communities [28,29,30,31,32]. Taken together, these interactions play a key role in the shaping of the microbiome. Meanwhile, correlations between bacteria in the vaginal communities during early pregnancy are not described. Here, we found 32 very strong correlations (R ≥ |0.8| and *p* ≤ 0.05) between bacteria in the cervical canal and 22 in cervicovaginal samples. The strongest correlations were found in non-abundant species. To our knowledge, there is no research explaining these relationships in the cervical or vaginal environment. Further in vitro and in vivo research are needed to determine the nature of these relationships and their role in pathological conditions.

## 5. Conclusions

The data we obtained demonstrate that *L. iners* is a predominant microorganism in Eastern European women’s cervical and cervicovaginal microbiome. 

The data on the cervical microbiome from the samples, taken by the cytobrush and therefore containing intracellular bacteria, were obtained for the first time. We found a higher abundance of *Muribaculum intestinale*, *Aquabacterium parvum* and *Methyloversatilis universalis* in the cervix compared to mixed cervicovaginal samples. The presence of *Muribaculum intestinale* and *Methyloversatilis universalis* in the female genital tract has never been described before. The soil bacteria *Psychrobacillus psychrodurans* has a higher abundance in the cervicovaginal samples compared to cervical samples, which can be explained by external contamination in the studied population from rural regions. 

There is almost no data about correlations between the presence of different species of bacteria in the female genital tract. Meanwhile, this might help to understand the mechanisms of the antagonistic and synergistic relationships between bacteria and their influence on microbiome formation. To our knowledge, we provided this analysis for the first time. We found 32 very strong correlations between bacteria in the cervical canal and 22 in cervicovaginal samples, including three complete direct correlations in the cervical microbiome and four in the cervicovaginal microbiome. The mechanisms of these relationships in the female genital tract and their contribution to pathological conditions remain unknown. 

In general, this study contributes to understanding cervical and vaginal microbiome composition in early pregnancy in different populations. It has been shown that cervical and cervicovaginal microbiomes in Eastern European women with ongoing pregnancies in the first trimester of pregnancy in all cases are *L. iners*-dominated and have very few differences between each other. Differences in the normal pregnancy microbiomes in different populations should be taken into account in multicentered microbiome research. Lifestyle, diet, level of urbanization, genetic factors and other causes should be the future subjects for the research to explain differences found in the genital tract microbiome in different populations.

## Figures and Tables

**Figure 1 microorganisms-10-02368-f001:**
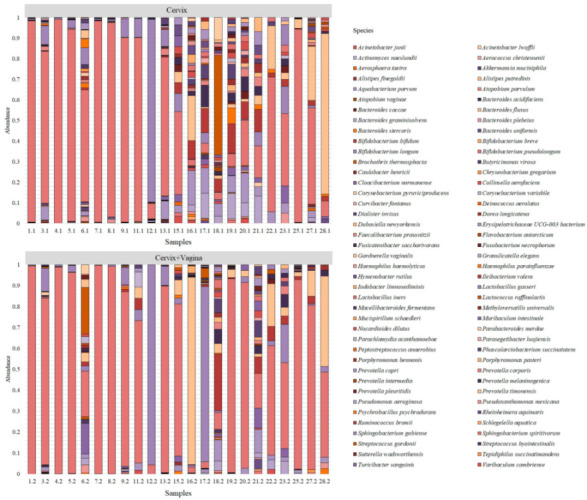
Bacterial species were detected in the studied samples using sequencing.

**Figure 2 microorganisms-10-02368-f002:**
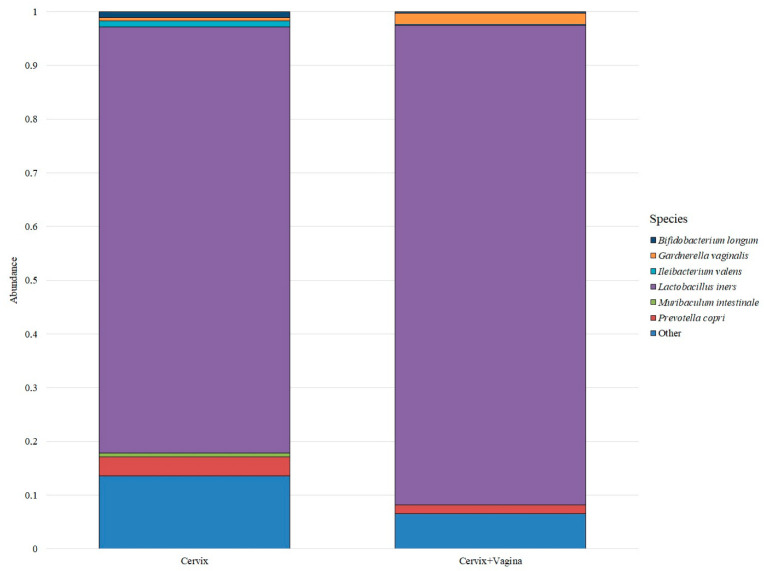
Bacterial species that form the microbiome core in both study groups.

**Figure 3 microorganisms-10-02368-f003:**
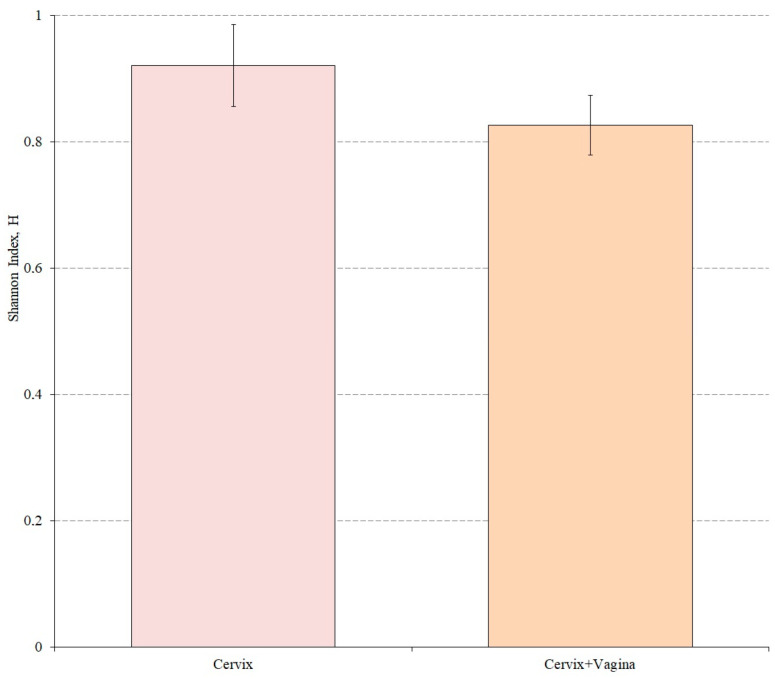
Shannon index as an indicator of alpha diversity for the studied groups (*p* = 0.0181).

**Figure 4 microorganisms-10-02368-f004:**
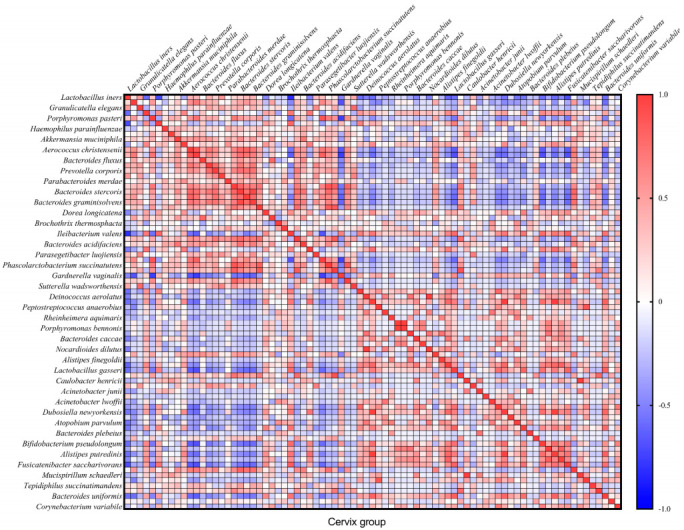
Correlation matrix describing the relationship between microorganisms in the cervical canal group.

**Figure 5 microorganisms-10-02368-f005:**
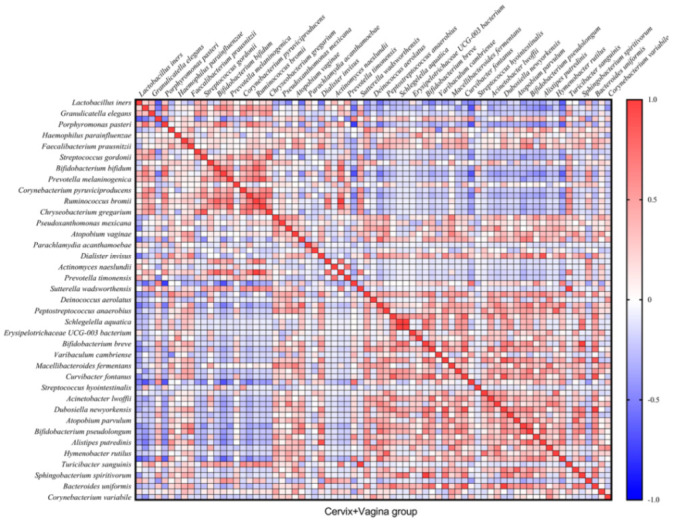
Correlation matrix describing the relationship between microorganisms in the cervical canal + vagina group.

**Table 1 microorganisms-10-02368-t001:** Clinical characteristics of patients enrolled in the study (*n* = 22).

Parameters	Mean ± SEM
Age (years)	31.68 ± 1.15
Gravidity	1.18 ± 0.37
Parity	1.18 ± 0.24
Number of miscarriages in anamnesis	0.27 ± 0.10
Number of artificial abortions in anamnesis	0.41 ± 0.14
Weight	67.38 ± 2.68
Height	165.68 ± 1.24
BMI	24.42 ± 0.76

## Data Availability

The row sequencing data are available in the NCBI BioProject database (BioProjectID: PRJNA886610, https://www.ncbi.nlm.nih.gov/bioproject/?term=PRJNA886610; accessed on 25 November 2022).

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
