# Peer review of "Lower Genital Tract Microbiome in Early Pregnancy in the Eastern European Population"

_microorganisms, 2022, doi:10.3390/microorganisms10122368_

Round 1
Reviewer 1 Report
The manuscript title “Lower genital tract microbiome in early pregnancy in Eastern European population” describe the microbiome present in lower genital tract of women in their first trimester of pregnancy.
This manuscript is generally well written but there are some major issues surrounding the microbiome analysis and interpretation of data as described below:
1. The general identification of the taxonomic composition of microorganisms was carried out based on the SILVA database version 132. SILVA package is used to assign Texas mostly in GI microbiome and 97% similarity can misrelate them to gut bacteria while KWON database or databases related to vaginal microbiome are more accurate in identifying vaginal microbiome. Using database related to vaginal microbiome will be important to identify vaginal microbiome.
2. Bifidobacterium longum and other bacteria belongs to human gastrointestinal tract. There are no reports of Pseudomonas aeruginosa found in female genital tract.
3. Ileibacterium valens and Muribaculum intestinale is from in murine intestine and cecum respectively. The author claim that this is the first paper to report this which makes me question is their cyto brushes were checked for DNA contamination?
4. Author didn’t perform correlation of microbiome and pregnancy outcome. The microbiome data was described as CST III and CST IV and highly diverse which shows dysbiotic condition in genital tract of these women. The connection to reproductive health and pregnancy outcome is missing which lessens the importance of this study.
Author Response
Please see the attachment.
Reviewer point #1.1: The general identification of the taxonomic composition of microorganisms was carried out based on the SILVA database version 132. SILVA package is used to assign Texas mostly in GI microbiome and 97% similarity can misrelate them to gut bacteria while KWON database or databases related to vaginal microbiome are more accurate in identifying vaginal microbiome. Using database related to vaginal microbiome will be important to identify vaginal microbiome.
Author response #1.1: Thank you for your comment. After careful analysis of your comment, we have found that we incorrectly specified the version of the SILVA database used. In our study, we used a newer version of 138 and not 132, as we wrote earlier. We have corrected the manuscript. Unfortunately, we were unable to find literature data on the limitations of the SILVA database for vaginal microbiome classification. It provides a comprehensive resource for up-to-date, quality-controlled databases of aligned ribosomal RNA (rRNA) gene sequences from the bacterial, archaean, and eukaryotic domains. In addition, it is SILVA that, compared to other widely used databases, contains the largest number of unique taxa included in the classification [Galloway-Peña J, Hanson B. Tools for Analysis of the Microbiome. Dig Dis Sci. 2020;65(3):674-685. doi:10.1007/s10620-020-06091-y; Yilmaz P, Parfrey LW, Yarza P, et al. The SILVA and "All-species Living Tree Project (LTP)" taxonomic frameworks. Nucleic Acids Res. 2014;42(Database issue):D643-D648. doi:10.1093/nar/gkt1209; Hugerth, L. W., Pereira, M., Zha, Y., Seifert, M., Kaldhusdal, V., Boulund, F., ... & Engstrand, L. (2020). Assessment of in vitro and in silico protocols for sequence-based characterization of the human vaginal microbiome. Msphere, 5(6), e00448-20.]. Also, the SILVA database is the main database, which is used in vaginal microbiome research, including the recent classification by France, Ravel et al. (2020), which is an upgraded version of Ravel’s classification (2011) - the most widely used classification of the vaginal microbiome. This classification, in particular, CST III and CST IV, was mentioned by you in the review (point 1.4). Thus, updated Ravel's classification of the vaginal microbiome is based on the SILVA database [France, M. T., Ma, B., Gajer, P., Brown, S., Humphrys, M. S., Holm, J. B., ... & Ravel, J. (2020). VALENCIA: a nearest centroid classification method for vaginal microbial communities based on composition. Microbiome, 8(1), 1-15.]. Moreover, there is a large number of other articles on the vaginal microbiome using the SILVA database, here are some examples [Van Der Pol, W. J., Kumar, R., Morrow, C. D., Blanchard, E. E., Taylor, C. M., Martin, D. H., ... & Muzny, C. A. (2019). In silico and experimental evaluation of primer sets for species-level resolution of the vaginal microbiota using 16S ribosomal RNA gene sequencing. The Journal of infectious diseases, 219(2), 305-314.; Becerra-Mojica, C. H., Parra-Saavedra, M. A., Diaz-Martinez, L. A., Martinez-Portilla, R. J., & Orozco, B. R. (2022). Cohort profile: Colombian Cohort for the Early Prediction of Preterm Birth (COLPRET): early prediction of preterm birth based on personal medical history, clinical characteristics, vaginal microbiome, biophysical characteristics of the cervix and maternal serum biochemical markers. BMJ open, 12(5), e060556.; Raimondi S. et al. Vaginal and Anal Microbiome during Chlamydia trachomatis Infections. Pathogens 2021, 10, 1347. – 2021.].
Reviewer point #1.2: Bifidobacterium longum and other bacteria belongs to human gastrointestinal tract. There are no reports of Pseudomonas aeruginosa found in female genital tract.
Author response #1.2: Bifidobacterium longum is a very important bacteria in the human body. Indeed, for the first time, this species was isolated from human faeces. However, this bacterial species was also detected in the vagina. Located in the gastrointestinal tract and vagina, Bifidobacterium longum has an impact on the digestive and immune systems. This species may also help prevent vaginitis and inflammation of the vagina because it can produce lactic acid [Korshunov VM, Gudieva ZA, Efimov BA, et al. The vaginal Bifidobacterium flora in women of reproductive age. Zh Mikrobiol Epidemiol Immunobiol. 1999;(4):74-78; De Seta F, Campisciano G, Zanotta N, Ricci G, Comar M. The Vaginal Community State Types Microbiome-Immune Network as Key Factor for Bacterial Vaginosis and Aerobic Vaginitis. Front Microbiol. 2019;10:2451. Published 2019 Oct 30. doi:10.3389/fmicb.2019.02451; Chaban B, Links MG, Jayaprakash TP, et al. Characterization of the vaginal microbiota of healthy Canadian women through the menstrual cycle. Microbiome. 2014;2:23. Published 2014 Jul 4. doi:10.1186/2049-2618-2-23; Freitas AC, Hill JE. Bifidobacteria isolated from vaginal and gut microbiomes are indistinguishable by comparative genomics. PLoS One. 2018;13(4):e0196290. Published 2018 Apr 23. doi:10.1371/journal.pone.0196290; Freitas, A. C., & Hill, J. E. (2017). Quantification, isolation and characterization of Bifidobacterium from the vaginal microbiomes of reproductive aged women. Anaerobe, 47, 145-156.]. Moreover, in a recent classification of the vaginal microbiome (VALENCIA) by France, Ravel et al. (2020), the Bifidobacterium-dominated microbiome was even classified as CST IV-C3 subtype of the vaginal microbiome [France, M. T., Ma, B., Gajer, P., Brown, S., Humphrys, M. S., Holm, J. B. & Ravel, J. (2020). VALENCIA: a nearest centroid classification method for vaginal microbial communities based on composition. Microbiome, 8(1), 1-15.]
In our study, the average abundance of Pseudomonas aeruginosa in the cervical canal group was 2.28%, and in the cervical canal+vagina group was 1.00%. So, this bacterium was found in 7 patients. Pseudomonas aeruginosa are opportunistic microorganisms that often cause nosocomial infections of various localizations, including the urogenital tract. This bacterium has previously been isolated from a vaginal swab as well as a catheterized urine sample. Pseudomonas aeruginosa in the cervical microbiome of non-pregnant women is associated with baseline HPV negativity. One study talks about pyometra caused by Pseudomonas aeruginosa infection. Thus, our study confirms the likelihood of localization of this bacteria in the female genital tract. [Johnson G, Mores CR, Wolfe AJ, Putonti C. Draft Genome Sequences of Two Pseudomonas aeruginosa Isolates from the Female Urogenital Tract. Microbiol Resour Announc. 2020;9(1):e01378-19. Published 2020 Jan 2. doi:10.1128/MRA.01378-19; McLeod N, Lastinger A. Pyometra due to Pseudomonas aeruginosa. IDCases. 2019;17:e00554. Published 2019 May 7. doi:10.1016/j.idcr.2019.e00554; Ritu, W., Enqi, W., Zheng, S., Wang, J., Ling, Y., & Wang, Y. (2019). Evaluation of the associations between cervical microbiota and HPV infection, clearance, and persistence in cytologically normal women. Cancer Prevention Research, 12(1), 43-56.] Also, Pseudomonas aeruginosa is a gram-negative bacteria, which is a well-known classical etiological factor of sepsis during pregnancy and puerperium due to its persistence in the genital tract [Majangara, R., Gidiri, M. F., & Chirenje, Z. M. (2018). Microbiology and clinical outcomes of puerperal sepsis: a prospective cohort study. Journal of Obstetrics and Gynaecology, 38(5), 635-641.; Bacterial sepsis in pregnancy: Green-top guideline №64а. Royal College of Obstetricians and Gynecologists. April 2012. 14 p.]
Reviewer point #1.3: Ileibacterium valens and Muribaculum intestinale is from in murine intestine and cecum respectively. The author claim that this is the first paper to report this which makes me question is their cyto brushes were checked for DNA contamination?
Author response #1.3: Thank you for your comment. Indeed, the presence of Ileibacterium valens and Muribaculum intestinale was an unexpected finding in our study. However, it should be noted that all cytobrushes for material sampling were sterile and packed in disposable bags, whose integrity was not compromised until the biomaterial was taken. In addition, the control sample required to remove contamination at the stage of data processing contained Milli-Q water and a cropped cytobrush lowered into it. This sample went through all the same stages of sample preparation as the samples under study. Therefore, we exclude the possibility of contamination. We have tried to describe this step in more detail in the methods section. We assume that the unexpected finding of Ileibacterium valens and Muribaculum intestinale in the studied samples may be due to patients living in rural areas and constant contact with wild and domestic animals. Necessary clarifications on the territory of residence of patients have been added to the “Materials and Methods” section.
Reviewer point #1.4: Author didn’t perform correlation of microbiome and pregnancy outcome. The microbiome data was described as CST III and CST IV and highly diverse which shows dysbiotic condition in genital tract of these women. The connection to reproductive health and pregnancy outcome is missing which lessens the importance of this study.
Author response #1.4: Our study included patients with normal pregnancy, delivered at term (37 to 42 weeks of gestation). In all patients, except for one, there was a predominance of Lactobacillus iners, which allowed us to conclude that they belonged to CST III. In the “Discussion” section we analyzed other research on the I-trimester microbiome, and in the majority of them, CST III was the predominant type of the microbiome with a further shift to other types of the microbiome in the II and III trimesters. This allows considering CST III as a normal type of microbiome during early pregnancy. Only in patient 28, we observed the predominance of Gardnerella vaginalis, which allowed us to refer her to the CST IV group. The patient did not find any abnormalities in other tests (see Supplementary Table 1), and there were no complaints during the history taking. In any case, the pregnancy ended in healthy delivery in all patients.

Reviewer 2 Report
In the article “Lower genital tract microbiome in early pregnancy in Eastern European population”, the authors describe the cervicovaginal microbiome of Eastern European women in the first trimester of pregnancy using the 16S rRNA sequencing method. The results obtained contribute to understanding the microbiome of pregnant women in Eastern Europe.
The article is descriptive and well written. However, the ethnic characteristics of the study population and their geographical distribution are not described. In the results and conclusions of the article, the authors refer to the fact that the women belong to rural areas, but this is not made clear or specified in the materials and methods. This information is important for the interpretation of the results obtained and should be incorporated into the article. The conclusion in the abstract does not highlight the main findings of the results obtained in this study and should be rewritten.
Author Response
Please see the attachment.
Reviewer point #2.1: In the results and conclusions of the article, the authors refer to the fact that the women belong to rural areas, but this is not made clear or specified in the materials and methods. This information is important for the interpretation of the results obtained and should be incorporated into the article.
Author response #2.1: Thank you for your comment. You are right, that territorial belonging plays a big role in the interpretation of the received data. We have added data to the “Materials and Methods” section regarding the ethnicity and the territory of residence of the patients.
Reviewer point #2.2: The conclusion in the abstract does not highlight the main findings of the results obtained in this study and should be rewritten.
Author response #2.2: Thank you for carefully reviewing our manuscript. We have rewritten the conclusion in the abstract. We hope that it now better images the conclusions we can make from the results of this research.

Reviewer 3 Report
This study aimed to characterize microbiome of the lower genital tract in Eastern European women during early pregnancy. Microbiome was evaluated at cervical and cervicovaginal level in women with first-trimester pregnancy, further evolved at term, by employing the 16S rRNA sequencing method. In both groups, the bacterial population located at the cervical level showed a prevalence of Lactobacillus iners while, at the cervicovaginal level, there were several other bacterial species including Gardnerella vaginalis and Prevotella copri. Moreover, the cervical microbiome evidenced higher alpha diversity with higher abundance of some bacteria like Muribaculum intestinale and lower abundance of Psychrobacillus psychrodurans. It was concluded that differences in the normal pregnancy microbiomes in various populations should be characterized also in relation to influencing variables such as urban and rural residence as well as soil and animal microbiota.
Introduction deals with the region- and race-dependent variability of microbiome also with reference to various biological factors.
Materials and Methods report clinical characteristics of the enrolled patients and describe accurately the methodologies that were employed for DNA extraction and 16S rRNA amplification and for library preparation and sequencing. In particular, the hypervariable region V3 of the 16S rRNA gene was chosen for the microbiome study. Processing of the sequencing data was carried out through elaborated bioinformatic analyses. Statistical analysis appears to have been suitably performed.
Results are exposed adequately and five figures efficaciously support the text.
Discussion underlines the originality of the present study also considering other comparable researches in this field carried out in various countries on pregnant women of different ethnic groups and under different anthropological conditions.
Overall, the present study may be considered somewhat original in contributing to extend, on an ethnical point of view referred to pregnant women, the knowledge on microbiome composition in the lower genital tract. The obtained results might help to also improve the choice of suitable drugs for treatment of infections at this level. The study was performed with methodological accuracy and technical competence and the manuscript was prepared with appreciable care. Sentence fluency, “English style”, lexicon and expositive clarity are acceptable.
Author Response
Please see the attachment.
Response to Reviewer #3: We would like to thank you for carefully reviewing our manuscript. It is very valuable for us to receive such positive feedback on our work. We tried to make some clarifications to the work, for example, we described the stage of sample preparation in more detail, and also added information about the territory of the patient's residence. We hope that these changes will significantly improve the quality of the manuscript.
